# Viral Nucleases from Herpesviruses and Coronavirus in Recombination and Proofreading: Potential Targets for Antiviral Drug Discovery

**DOI:** 10.3390/v14071557

**Published:** 2022-07-16

**Authors:** Lee R. Wright, Dennis L. Wright, Sandra K. Weller

**Affiliations:** 1Department of Pharmaceutical Sciences, University of Connecticut School of Pharmacy, Storrs, CT 06269, USA; lee.wright@uconn.edu (L.R.W.); dennis.wright@uconn.edu (D.L.W.); 2Department of Molecular Biology and Biophysics, University of Connecticut School of Medicine, 263 Farmington Ave., Farmington, CT 06030, USA

**Keywords:** exonucleases, nucleases, herpesvirus, coronavirus, recombination, proofreading, antiviral, ExoN, alkaline nuclease

## Abstract

In this review, we explore recombination in two very different virus families that have become major threats to human health. The Herpesviridae are a large family of pathogenic double-stranded DNA viruses involved in a range of diseases affecting both people and animals. Coronaviridae are positive-strand RNA viruses (CoVs) that have also become major threats to global health and economic stability, especially in the last two decades. Despite many differences, such as the make-up of their genetic material (DNA vs. RNA) and overall mechanisms of genome replication, both human herpes viruses (HHVs) and CoVs have evolved to rely heavily on recombination for viral genome replication, adaptation to new hosts and evasion of host immune regulation. In this review, we will focus on the roles of three viral exonucleases: two HHV exonucleases (alkaline nuclease and PolExo) and one CoV exonuclease (ExoN). We will review the roles of these three nucleases in their respective life cycles and discuss the state of drug discovery efforts against these targets.

## 1. Introduction

It has long been recognized that genetic recombination plays many important roles in the biology of all living organisms, including the repair of double-strand breaks (DSBs) and the generation of diversity during evolution. Recombination not only allows cells to maintain chromosomal stability and prevent genetic loss but also enables organisms to adapt and evolve. Extensive research on the mechanisms of recombination in both prokaryotic and eukaryotic cellular organisms has revealed a complex set of pathways by which genetic information is exchanged and repaired. Given the importance of these processes, it is perhaps not surprising that viruses have also evolved to rely heavily on recombination for genome replication and repair and to promote viral diversity [1,2,3,4,5,6]. Although most DNA and RNA viruses utilize recombination during replication, in this paper, we focus on Herpes and Coronaviruses, as these two viruses, in particular, have evolved efficient recombination pathways that specifically utilize virally encoded exonucleases. Other viruses with high recombination rates utilize different mechanisms. For instance, influenza viruses have segmented genomes that can reassort when multiple strains coinfect cells, effectively generating novel subtypes and high levels of antigenic drift [7].

## 2. General Mechanism of Recombination

When double-strand breaks arise in cellular organisms, two major pathways of homology-driven DNA repair exist to repair broken ends and prevent genetic loss. Homologous recombination (HR) utilizes a classical recombinase that can carry out strand invasion (RecA for bacteria and Rad51 for eukaryotes) [8]. In the HR pathway, the repair of DNA DSBs is initiated by resection of a DSB by a combination of endo- and exonucleases (DNA2, EXO1 and MRE11) [8,9]. Resection generates single strand DNA (ssDNA) with a 3′ terminus that is coated by RAD51, and the RAD51-coated DNA initiates strand invasion in an ATP-dependent fashion [8]. The second pathway of homology-directed repair is single-strand annealing (abbreviated SSA). This pathway does not require strand invasion or ATP hydrolysis; however, it does require exonucleolytic resection and an annealing protein, such as RAD52 [10]. In cellular organisms, the HR pathway is the predominant homology-driven mode of recombination, and SSA is only activated under conditions in which the HR pathway has been inactivated [11]; however, for many DNA viruses, SSA appears to be the predominant mechanism for viral recombination. In fact, DNA viruses of bacteria, protozoa, plants, insects and mammals have now been shown to encode an evolutionarily conserved two-subunit recombinase that promotes single-strand annealing (SSA). The Exo/SSAP complexes are composed of a 5′-to-3′ exonuclease capable of catalyzing end resection and a single-strand annealing protein (SSAP) [4,5,12,13,14,15,16]. In this review, we discuss evidence that viral SSA pathways are essential not only for recombination but for viral replication as well. Although some viruses with small circular genomes, such as the papovaviruses, have evolved to utilize host DNA replication machinery to replicate their genomes [17], larger DNA viruses have evolved to Recombination-Dependent Replication (RDR) during genome replication using virally encoded recombinases [5]. The best-characterized example of an exo/SSAP complex is phage λ Redα/β, which in addition to its role in viral replication, has been used to promote in vivo recombination-mediated genetic engineering or recombineering [2,18,19]. Interestingly, herpes simplex virus (HSV) also encodes an exo/SSAP comprised of the 5′-to-3′ alkaline exonuclease (UL12) and an ssDNA annealing protein (ICP8), and our laboratory has shown that both activities are essential for HSV DNA replication and the formation of replication compartments [4,6,20,21]. Several of the DNA viruses that encode an Exo/SSAP have been shown to exhibit high rates of recombination, including human herpesviruses (HHVs), baculoviruses and lambda phage [5,16].

## 3. Human Herpesviruses (HHVs)

The *Herpesviridae* are a large family of pathogenic double-stranded DNA viruses involved in a range of diseases affecting both people and animals. Over 90% of the human population is infected with one or more of the nine human HHVs [22]. The HHV family is divided into three subfamilies: alpha (α), beta (β) and gamma (γ). Herpes simplex viruses 1 and 2 (HSV1/2) and varicella zoster virus (VZV) are α-HHVs; human cytomegalovirus (CMV), human herpesviruses HHV−6A, HHV-6B and 7 are β-HHVs; and Epstein–Barr Virus (EBV) and human herpesvirus 8 (Kaposi’s sarcoma-associated herpesvirus, KSHV) are γ-HHVs.

### 3.1. Primary Infection

Primary infection with Herpes simplex virus (HSV 1/2) is associated with painful blisters affecting the skin, mouth, lips, eyes and genitals, as well as life-threatening diseases such as encephalitis, meningitis and neonatal herpes. Primary infection with VZV causes varicella (chickenpox), which can be severe in immunocompromised individuals. Infections with CMV (a β-HHV) are often asymptomatic in immunocompetent hosts, although pregnant women can transmit the virus to a developing fetus, resulting in congenital infections that can lead to permanent hearing loss, loss of vision and/or mental impairment in infants and children [23,24]. During primary infection, CMV replicates in fibroblasts and epithelial cells. In immunocompromised hosts, CMV causes serious diseases such as retinitis, pneumonitis, myelosuppression and encephalitis [25,26]. HHV-6A, HHV-6B and HHV-7, known as human roseolaviruses, are associated with fever, rash and seizures, especially in children under the age of two. Although these viruses were first identified as lymphotropic viruses infecting T cells, it is now clear that they are neurotropic and can enter the central nervous system (CNS) during primary infection [27,28,29]. While α- and β-HHVs start their infection cycles as lytic infections, infection with the γ-HHVs (EBV and KSHV) first establish latent infection in B cells, which can later be reactivated. Primary infection with EBV causes infectious mononucleosis.

### 3.2. Latent Infection

All HHVs establish life-long, latent infections that undergo periodic reactivation in the host [30]. Reactivation of latent virus results in shedding of infectious virus in saliva and genital secretions. Immunocompromised patients, particularly transplant recipients, are at high risk for reactivation of diseases associated with all the HHVs. HSV1/2 establish latency in sensory neurons (trigeminal or sacral ganglia), and shedding from both symptomatic and asymptomatic individuals can lead to transmission of virus to vulnerable newborns as well as sexual partners. HSV1/2 reactivation can also lead to infection of the central nervous system. Reactivation of VZV leads to shingles. The β-HHVs (CMV, HHV-6A/B and HHV-7) establish latency in mononuclear cells as well as the CNS. CMV reactivation in transplant recipients is especially detrimental due to the cytopathic effects of the virus on host organ systems that can lead to life-threatening conditions such as retinitis, pneumonitis, myelosuppression, encephalitis and graft-versus-host disease [25,26,30,31,32,33]. HHV-6A/B and 7 reactivation is associated with fever, hepatitis and encephalitis, as well as higher rates of HCV progression and CMV reactivation, graft failure and mortality [34,35]. The human γ-HHVs, EBV and Kaposi’s sarcoma–associated herpesvirus (KSHV), establish latency in B cells and are associated with several B cell and epithelial cell malignancies in which tumor cells are latently infected with EBV or KSHV. These malignancies include Hodgkin’s lymphoma, Burkitt’s lymphoma, gastric carcinoma and nasopharyngeal carcinoma associated with EBV, and primary effusion lymphoma, multicentric Castleman disease and Kaposi’s sarcoma associated with KSHV. The number of patients experiencing serious HHV-related problems is increasing as the number of individuals undergoing hematopoietic stem cell transplants (HSCT) or receiving immunosuppressive chemotherapy grows. Among the most problematic HHVs in the transplant setting are the α-herpesviruses (HSV-1/2) and β-herpesviruses (CMV and HHV-6B) [34,35,36,37].

### 3.3. Possible Association of HHVs with Neurodegenerative Diseases

HHVs are known to induce inflammation, especially in neuronal tissues, leading to serious neurological diseases such as encephalitis, meningitis and epilepsy. HSV, CMV and the roseolaviruses have been shown to infect the central nervous system (CNS), possibly due to the ability of these viruses to alter the permeability of the blood–brain barrier. A growing body of evidence suggests that human HHVs are linked to degenerative central nervous system conditions such as Alzheimer’s disease (AD) and multiple sclerosis (MS) [38,39,40,41,42,43,44,45]. The brains of individuals with AD have been reported to contain RNA from HSV, HHV-6 and HHV-7 [46]. AD is known to be associated with cerebral aggregation of the β-amyloid peptide (Aβ), and it is often suggested that these aggregates induce AD. However, recent studies suggest that peptides such as Aβ may actually be part of an ancient innate immune response that cells use to defend themselves against infection. This model, known as the antimicrobial protection hypothesis [47,48], posits that Aβ is used to entrap and neutralize invading pathogens in β-amyloid fibrils. Fibrillization induces neuroinflammatory pathways that help fight the infection. However, in AD, it has been suggested that chronic activation of this pathway leads to sustained inflammation, aggregation of amyloid in the brain and neurodegeneration. Thus, Aβ aggregation may be part of a normal response to infection that progresses to a dysregulated response when it cannot be cleared. Infection by several types of pathogens, including bacteria and other viruses, could be responsible for the induction of antimicrobial peptides, and a preponderance of the data published to date support the notion that the HHVs could play a role in AD and perhaps other neurodegenerative diseases. Interestingly, a comprehensive longitudinal study recently published implicated the γ-HHV, EBV, as a leading cause of multiple sclerosis [49]. One important implication of the antimicrobial protection hypothesis is that it may be possible to slow or prevent the onset of AD through the use of antiviral agents that inhibit HHV infections; indeed, several recent reports support this contention [50,51].

### 3.4. Replication Strategies of the HHVs

Although the α-, β- and γ-HHVs differ in their tissue tropisms and associated pathologies, most of the replication machinery is conserved across all three subfamilies, suggesting that the mechanisms by which they replicate their DNA genomes are similar [3]. Studies on HHV DNA replication have been primarily performed on herpes simplex virus due to its amenability to genetic and biochemical analysis [3], and we will, therefore, focus our attention on the replication strategy of HSV. Studies from our lab have revealed that the HSV replication machinery promotes a unique form of DNA replication that utilizes a recombination-dependent mechanism to produce concatemers, which are required for packaging infectious virus [4,5,6,20,21,52,53,54]. Seven viral proteins have been identified as essential for HSV DNA synthesis: UL9 (origin-binding protein/helicase), UL5/UL8/UL52 (helicase/primase), ICP8 (single-strand annealing protein) and UL30/UL42 (HSV Pol). UL30 is a B-subfamily DNA polymerase and is comprised of two functional domains: a 3′-to-5′ exonuclease, PolExo, that plays a role in proofreading and the catalytic polymerase domain required for extending primers during viral DNA replication [55,56,57,58,59]. The PolExo will be described below as a potential target for antiviral therapy.

Several lines of evidence support the notion that HSV performs recombination-dependent replication. Evidence for high rates of recombination between coinfecting HSV genomes comes from cell culture and animal infection models, as well as from analysis of viruses circulating in patient populations [60,61,62,63,64]. We and others have reported that viral replication intermediates are composed of complex X- and Y-branched structures as evidenced by electron microscopy [65,66] and pulsed-field gel electrophoresis [67,68]. We have shown that HSV alkaline nuclease, UL12, and ICP8 function together as a two-subunit recombinase (Exo/SSAP) that can carry out strand exchange and stimulate SSA [4,14,15]. Our model for HSV DNA replication is that this complex (UL12/ICP8) promotes a series of reactions in which UL12 resects dsDNA, leaving a 3′ ssDNA overhang that is recognized by ICP8. ICP8 then promotes annealing of the ssDNA to an active replication fork to promote DNA synthesis by the viral DNA polymerase, leading to the formation of concatemeric replication intermediates [5,6,20]. The exonuclease activity of UL12 and the annealing function of ICP8 are essential for HSV replication [4,6,20]. Several of the DNA viruses that encode an Exo/SSAP have been shown to exhibit high rates of recombination, including HHVs, baculoviruses and lambda phage [5,16]. The essentiality of the exonuclease activity for the production of infectious HSV indicates that this viral nuclease will be a valuable target for the development of novel antivirals [20].

### 3.5. Current Standard of Care

Currently, HHV infections are treated primarily with agents that target viral DNA polymerases. These include nucleoside analogues acyclovir/ganciclovir (ACV/GCV), nucleotide analogues (cidofovir) and pyrophosphate mimetics (foscarnet). ACV/Val-ACV is used for first line HSV therapy and prophylaxis, although long-term treatment can lead to the development of drug resistance, especially in immunocompromised patients [69,70]. GCV is used for CMV therapy and prophylaxis; however, it causes significant myelosuppression and toxicity to hematological cells, a particularly devastating side effect in allogeneic HSCT patients. Even brincidofovir, another polymerase inhibitor designed to decrease renal toxicity, is associated with diarrhea, acute GVHD and adverse gastrointestinal events [71]. In addition to these safety issues, nucleoside analogues have a narrow spectrum owing to the need for bioactivation. A recently approved CMV-specific drug, letermovir, utilizes a different mode of action [72]. Instead of targeting the viral polymerase, it inhibits the viral terminase, an enzyme required for packaging the viral genome. Despite its promise as a new antiviral agent with a novel mode of action, letermovir appears to exhibit rapid onset of resistance [73,74,75,76] and drug–drug interactions with post-transplant immunosuppressive medications [77]. Recently, a new CMV-specific antiviral agent, maribavir, gained FDA approval [78]. This agent specifically targets the UL97 kinase, which is required for DNA replication and nuclear egress [79]. An additional agent with a novel target, HSV helicase–primase, is currently in clinical trials [80]. A comprehensive list of drugs targeting HHVs can be found in Appendix A. Thus, the agents currently in use for HHVs are associated with dose-limiting toxicity and/or narrow antiviral spectrum. In summary, there is a compelling need for safe, effective agents that utilize novel modes of action. New therapeutics would not only be important for the treatment of resistant viruses but also for use in combination therapy to lower dose-limiting toxicities. Indeed, such combinations could prove useful in preventing the spread and pathogenicity of HHVs. In addition, new modalities of anti-HHV therapy may be expected to delay onset or prevent neurodegenerative disease.

## 4. Introduction to Coronaviruses

Coronaviruses (CoVs) are enveloped viruses containing a single-strand, positive-sense RNA genome of approximately 26–32 kilobases, an unusually large size for an RNA virus [81]. CoVs can be classified into four genera: the α-, β-, γ- and d-CoVs; however, only α- and β-CoVs can infect mammals, while γ-CoVs infect avian species, and d-CoVs can infect both mammals and avian species [82]. The circulating α-CoVs (*HCoV-229E*, *HCoV-OC43*, *HCoV-NL63* and *HKU1*) are responsible for ~15–30% of cases of the seasonal “common cold” in humans and are largely associated with relatively mild symptoms [83,84]. Other zoonotic CoVs, which mainly circulate in lower animals, have demonstrated a propensity to “leap” to human hosts [85]. Over the past two decades, three β-coronaviruses (CoVs), have emerged as major threats to the human population, causing severe respiratory and other infections: *SARS-CoV-1* (2002–2004) [86], Middle East respiratory syndrome *MERS-CoV* (2012–present) [87,88] and *SARS-CoV-2* (2019–present) [89]. Such spillover events typically require an additional “hop” from their primary reservoirs (e.g., horseshoe bat) to an intermediate reservoir species (e.g., civet cats, camels or pangolins) before infecting humans [90,91,92,93,94]. The rapid evolution of novel chimeric genomes with the continued emergence of additional variants with altered host range and tissue tropism has captured the world’s attention and left researchers scrambling to understand the mechanisms behind this phenomenon. The trailblazing work by pioneering CoV researchers Michael Lai, Ralph Baric, Mark Dennison, Stanley Perlman and others has elucidated the critical role of recombination in the CoV life cycle. Recently, the Denison group made the remarkable discovery that an unusual CoV protein, the exoribonuclease (nsp14/ExoN), is essential for recombination in CoVs [95], discussed in further detail below. Even before the current pandemic, it was known that the proofreading protein ExoN was critically involved in RNA synthesis, replication fidelity, fitness, ribavirin resistance and evasion of cellular immune responses [96,97,98,99,100,101,102,103,104]. ExoN will be a primary focus in this review article, and we will discuss possible strategies and benefits to targeting ExoN for drug discovery.

### 4.1. Role of Recombination in CoV Infection

It has been recognized since the early 1960s that RNA–RNA genome recombination can occur in RNA viruses such as poliovirus [105,106]. Recombination has been observed in several RNA viruses both in vitro and in vivo [107,108,109,110,111] and is generally thought to occur by polymerase jumping (or copy choice) during RNA synthesis [108]. Copy-choice RNA recombination occurs when partially synthesized RNAs dissociate from one template and then rejoin the same or another template followed by elongation. Although this phenomenon has been recognized in many RNA viruses, the rates of CoV recombination are much higher than other RNA viruses [110,112]. As described below, the higher rates of recombination associated with the CoV replication/transcription machinery is dependent on ExoN, a viral exonuclease [95]. Strikingly, ExoN is required not only for proofreading during viral replication but also for the high rates of recombination observed in this family of viruses [95]. The high propensity for recombination in CoVs is not only important for the emergence of novel strains, but it also explains many aspects of the pathogenesis of CoV infections. For instance, recombination in zoonotic CoVs has been implicated in spread and severe disease in livestock animals, resulting in vaccine failure in pigs and chickens [113,114]. Recombination is now well-recognized in the evolution of the pathogenic human CoVs SARS1/2 and MERs from zoonotic CoVs [89,91,92,115,116,117]. The continued evolution of new variants of *SARS-CoV-2* in patient samples [118] also relies on recombination and has important implications for our ability to monitor the virus using current testing protocols and, more importantly, threatens to undermine the efficacy of current CoV vaccines. Understanding these mechanisms will be crucial to our ability to treat and control this and future CoV pandemics.

### 4.2. General Aspects of CoV Replication

After fusion of the enveloped CoV virus particle at the cell membrane, the capped positive-strand genome is deposited into the cytoplasm where it can be directly translated to produce two large overlapping ORFs (*ORF1a* and *ORF1b*) that are subsequently, proteolytically processed to generate 16 non-structural proteins (nsps) (nsps1–16) [119,120]. Nsp7–16 make up the CoV replicase complex that carries out replication and transcription, referred to as the replication/transcription complex or RTC. The RTC performs two essential functions: the replication of genomic RNA and the production of capped mRNAs using replicase proteins. These steps include the replication of the parental (+) strand to form a (−) RNA strand. The (−) strand can subsequently be used to generate additional (+) stranded viral genomes and for the transcription of subgenomic mRNAs that encode viral structural proteins. The subgenomic mRNAs are generated by a discontinuous viral transcription process, producing a set of nested 3′ and 5′ co-terminal subgenomic RNAs (sgRNAs) [121,122,123]. The RTC associates with modified cellular membranes to form replication factories, which provide a favorable microenvironment for replication and transcription [124].

### 4.3. CoV Replicase Proteins

Several of the CoV replicase proteins are universal among positive-strand RNA viruses, such as RNA-dependent RNA polymerase (RdRp, nsp12) and helicase (nsp13); however, in addition, CoVs and the other Nidoviruses encode proteins that are unique and play roles not typically associated with RNA viruses [86,125,126,127,128]. Viral RdRps exhibit high mutation rates due to the lack of proofreading activity, and high mutation rates are believed to restrain the size of viral genomes [129]. As a result, most RNA virus genomes are less than 15 kb. Interestingly, the CoV genome size is quite large (~30 kb) [127]. In order to replicate a genome of this size, CoVs have made at least two important evolutionary adaptations. One is that the CoV RdRp has acquired a processivity factor, comprised of scaffold proteins nsp7 and nsp8. The nsp12–nsp7–nsp8 complex is the minimal complex required for nucleotide polymerization [130], and the presence of the nsp7/8 scaffold contributes to fidelity of RNA synthesis [131] as well as to the ability to replicate long genomes [132].

The second evolutionary adaptation made by CoVs to improve replication fidelity is the acquisition of a novel RNA exoribonuclease (ExoN) with proofreading ability [98]. Proofreading is mediated by the 3′-to-5′ exoribonuclease activity of ExoN and involves the removal of misincorporated bases, thus ensuring replication fidelity. ExoN is encoded by nsp14, a bifunctional protein with two domains: a 3′-to-5′ exoribonuclease (ExoN) and a methyltransferase (N7-MTase) [133,134]. Nsp14 also interacts with nsp10, a cofactor that strongly enhances ExoN activity [133,135,136,137,138]. The importance of this proofreading activity was demonstrated by the Denison lab, showing that exoribonuclease-deficient CoV mutants demonstrate impaired RNA synthesis and high levels of mutation [97,99,100,101,103].

An interesting result of the acquisition of ExoN is its effect on the sensitivity of CoVs to nucleoside analogue inhibitors. Although many RNA viruses can be treated with ribavirin, this drug has minimal activity against the CoVs. Ribavirin is thought to be misincorporated into the RNA of sensitive viruses, such as RSV and HCV, resulting in lethal mutagenesis in the viral progeny [133]. ExoN is responsible for resistance to nucleoside analogues such as ribavirin by excising the drug from the growing RNA chain, allowing normal replication to proceed. Exoribonuclease-deficient mutants are significantly more sensitive to ribavirin and remdesivir, consistent with ExoN’s critical role in resistance to mutagenic nucleoside analogues [101,133,139].

### 4.4. Identification of ExoN as a Driver for Recombination

As previously noted, ExoN has been shown to be essential for recombination in three different strains of CoV, including murine hepatitis virus (MHV) and the β-coronaviruses SARS-CoV-2 and MERS-CoV, as demonstrated by RNA next-generation sequencing technology [95]. All three of these viruses were shown to be capable of generating recombination products during replication in culture. The depletion of nsp14-ExoN activity in MHV leads to an alteration in recombination patterns and decreased recombination frequencies [95]. The production of subgenomic mRNA transcripts is also decreased in ExoN mutants, suggesting that nsp14/ExoN is required for both recombination and discontinuous synthesis of subgenomic RNAs. Thus, ExoN is essential for several aspects of the CoV life cycle, including RNA replication/transcription/fidelity, recombination, fitness and ribavirin resistance, as well as evasion of cellular immune responses [84,95,99,128]. Interestingly, in some ways, CoVs more closely resemble the replication machinery of DNA viruses such as the HHVs [128,140].

### 4.5. Current Standard of Care

While multiple coronavirus vaccines were rapidly developed in an effort to reduce the uncontrolled spread of *SARS-CoV-2*, evasive variants quickly appeared, underscoring the fact that vaccines alone are not sufficient to combat the current crisis and should not be solely relied upon to prepare for future emergences. Currently, there are limited treatment options for CoV infections beyond the aforementioned mRNA vaccines. Several monoclonal antibody cocktails have proven effective; however, resistance of the newer variants to the mAb therapies is of concern. Historically, nucleoside and nucleotide analogues that target viral polymerases have provided an effective first-line therapy for RNA and DNA viruses alike. However, as described above, ribavirin and other nucleoside analogues are largely ineffective against CoVs because of ExoN [133]. The nucleoside analogue remdesivir (RDV) has been shown to have modest antiviral activity against *SARS-CoV*, *SARS-CoV-2* and *MERS-CoV* and has been approved for *SARS-CoV-2* use [141]. However, the necessity of administering RDV intravenously limits its use, thus its overall impact, to hospitalized patients [141]. Other therapeutics of note include ritonavir-boosted nirmatrelvir (Paxlovid) and molnupiravir, both of which have received Emergency Use Authorizations from the FDA for the treatment of COVID-19 [142,143]. Unfortunately, recent reports of high levels of “paxlovid rebound” raise concerns about the recommended dosing strategy [143]. It is crucial to explore new targets and approaches and expand the pipeline of therapeutics to protect against current and future biological threats from CoVs; this has now become a national and international priority. New direct-acting antivirals (DAAs) can also be given in combination therapy to lower dose-limiting toxicities and improve efficacy of nucleoside analogues.

### 4.6. Similarities between HHVs and CoVs

Although HHVs and CoVs differ in many respects, we are struck by some remarkable similarities. Recombination plays a major role in the replication strategies of both virus families, and the HHV and CoV nucleases involved in resection (alkaline nuclease and ExoN, respectively) are essential for viral replication. Both viruses encode proofreading exonucleases (PolExo and ExoN) that have been shown to be essential for ensuring replication fidelity. The evidence reviewed so far thus suggests that the three viral nucleases discussed herein provide promising new targets for antiviral therapy.

While a detailed discussion is beyond the scope of this review, it is worth noting that other viruses also stimulate recombination-dependent replication (RDR). Like the herpesviruses, baculoviruses encode a 5′-to-3′ exonuclease that interacts with an ssDNA annealing protein to make a complex (Exo/SSAP) that can stimulate SSA [16,144]. Interestingly, *vaccinia virus*, another family of large DNA viruses, employs an unusual recombination system in which the 3′-to-5′ exo of the viral polymerase is proposed to resect broken DNAs, generating 5′ ssDNA tails that anneal with complementary ssDNA [145,146]. The exonuclease involved in pox replication, which resides on the viral polymerase, is also responsible for proofreading and is analogous to the PolExo of the HHV DNA polymerase. It will be of interest to determine whether the HHVs also utilize PolExo to stimulate recombination, in addition to using the alkaline nuclease for this function.

## 5. Introduction to TMID Nucleases

As previously discussed, antiviral drug discovery has centered mainly around the misincorporation of toxic nucleotide/side analogues into the viral genome via the polymerase. The complex viral replication machinery of HHVs and CoVs offers a wealth of underexplored, essential targets involved in the synthesis and processing of genomic material. In addition to potential standalone antiviral agents, exploiting these essential targets would open the door for powerful combination therapies that could reduce the onset of resistance, improve efficacy and also aid in reducing dose-limiting toxicity.

The wide array of nucleic acid processing enzymes (nucleases), characterized by their ability to cleave the phosphodiester bonds of either DNA or RNA, has been extensively reviewed in [147]. This broad category can be conveniently subdivided into metal-dependent and metal-independent catalysis. Most metal-dependent nucleases utilize two metal ions, often divalent magnesium, to position and activate the phosphodiester bond for cleavage. These two-metal ion-dependent enzymes, referred to as TMIDs, perform critical functions in genome production, nucleic acid metabolism and proofreading. The active sites of TMID nucleases contain clusters of conserved carboxylates (i.e., DDE or DEDD motifs), which coordinate the divalent cations essential for substrate binding and catalysis [147]. Despite the similarities in the active sites, TMID nucleases are diverse with respect to function and substrate selectivity. Indeed, TMID nucleases can act as exo- or endonucleases, on DNA or RNA, and in a 5′-to-3′ or 3′-to-5′ direction [147].

Several notable efforts aimed at inhibiting enzymes of this type have produced novel DAAs. For example, compounds such as raltegravir and dolutegravir target the HIV integrase, a TMID nuclease that catalyzes an essential strand transfer reaction [148,149,150]. Another, more recent example is the antiviral baloxavir, which inhibits the influenza “cap-snatching” polymerase PA endoribonuclease subunit [151]. These novel approaches have changed the treatment paradigms for the diseases associated with both pathogens and demonstrated that targeting nucleases is a powerful antiviral strategy. The tremendous success with integrase and PA, both of which function as endonucleases, begs the question of whether similar approaches would be valuable in attacking viral exonuclease functions. Of particular interest to our lab are virally encoded exonucleases from HHV and CoV.

Exonucleases non-specifically cleave the terminal phosphodiester bond of either DNA or RNA, whereas endonucleases catalyze internal cleavage often in a sequence-selective manner. It is important to note that the exonucleases operate with a preferred polarity, excising specifically at either the 5′ or 3′ terminus. Mechanistically, TMID exonucleases employ the magnesium ions to both position and activate the phosphodiester bond for cleavage [147]. Bond scission is commonly mediated through direct attack of the phosphodiester by a nucleophilic water molecule, typically producing a 5′-phosphorylated mononucleotide (Figure 1).

## 6. Exonuclease Enzymes of HHV

Human herpes viruses encode several TMID nuclease enzymes from different superfamilies: the alkaline nuclease (DEK family) [152], the terminase (RNHL family) [153] and PolExo [55,56,57,59], which is also an RNHL family member [154]. Both the alkaline nucleases and pol/exo function as exonucleases, while the terminases are endonucleases and will not be discussed further in this review.

### 6.1. Alkaline Nuclease

All HHVs encode a well-conserved version of this important protein [155,156,157,158]. Sequence analysis of the AN proteins reveals high levels of conservation, particularly in the active site, among the HHV enzymes [159] (Figure 2). The AN proteins from HSV (UL12), CMV (UL98) and EBV (BGLF5) have been shown to be essential for virus growth [20,68,159,160,161,162,163]. Interestingly, the high degree of similarity among the HHV AN proteins allows CMV UL98 to functionally substitute for UL12 in cells infected with a UL12-null mutant virus [161,164], consistent with functional conservation.

Crystal structures of the alkaline nucleases from both EBV (HHV-4, BGLF5 apo PDB:2WB4) [163,165,166] and KSHV (HHV-8, SOX complex PDB:3POV) [167,168] have been solved and provide significant insight into the function and structure of these proteins. The alkaline exonuclease orthologs of all nine HHVs are highly homologous (Figure 3), containing a mixed beta sheet comprised of four strands sandwiched between two alpha helices. This forms the catalytic site containing the EDEK motif. All four of the EDEK residues coordinate one of the two central magnesium ions, while the other magnesium ion appears to create a bridge between the more superficial glutamate 184 (SOX) and the substrate. In addition to its association with the interior magnesium atom, the lysine coordinates a conserved water molecule that is ostensibly used to cleave the phosphodiester bond [167].

### 6.2. PolExo

The polymerase activity is a well-established antiviral target, primarily using nucleoside/tide inhibitors that are misincorporated during DNA synthesis [169,170,171,172]. The HHV polymerases also contains a second catalytic domain, a 3′→5′ proofreading exonuclease (PolExo) [55,56,57,58,59]. The wild-type HSV polymerase with the exonuclease domain was crystallized in the apo form several years ago (PDB ID: 2GV9) [56]. Based on the crystal structure of the PolExo domain and similarities with the bacteriophage RB69, residues D368, I369 (backbone), E370 and D471, along with D581, are predicted to be important for the coordination of the two Mg^2+^ ions required for exonuclease activity [56,57], the hallmark of a TMID enzyme. This putative active site also contains K539 bound to a conserved water molecule, likely critical for exonuclease activity. These key residues are identical in all nine of the HHV PolExos (Figure 4). PolExo activity of the HSV and CMV polymerase is essential for the fidelity of viral DNA replication [173,174,175,176].

### 6.3. Inhibition of Alkaline Nuclease and Pol/Exo Function

The genetic experiments described above support the hypothesis that AN is an excellent target for anti-HHV drug discovery. This hypothesis is also supported by previous efforts to inhibit ANs from HSV and CMV with small molecules [20,177,178,179] (Figure 5). A plant anthraquinone, emodin, was shown to inhibit the nuclease activity of HSV-1 UL12 in a biochemical assay. Emodin reduced plaque formation and induced the accumulation of viral nucleocapsids in the nucleus [177], consistent with our previous analysis of the phenotype of the UL12-null mutant, AN-1 [68,159,160,180]. Alam et al. reported that atanyl blue PRL inhibited the nuclease activity of purified CMV UL98 protein and inhibited viral spread [178], again consistent with our findings in HSV [20]. More recently, metal-directed hydroxytropolones such as compound 1, synthesized by the Morelli group, were demonstrated to suppress HSV infection [181,182], and we have shown that these compounds inhibited the AN activity of UL12 [20].

In our laboratory, we conducted high-throughput screening of a small library of compounds against the alkaline nucleases derived from SOX, EBV, CMV and HSV. This screen identified several hits in the low micromolar range, such as purpurogallin and compound 2 (Figure 5); however, these inhibitors are considered promiscuous metal-chelating agents with low selectivity, making them unattractive for further development. In a parallel effort, we created a privileged library of 250 compounds containing (a) known TMID-directed drug inhibitors of HIV integrase and influenza endoribonuclease, (b) compounds reported to inhibit TMID enzymes (tropolones) and (c) compounds predicted by flexible docking to bind to the active site of the KSHV SOX protein [183]. These compounds were screened for anti-nuclease activity against several known and potential TMID enzymes [183]. The results of the enzyme inhibition assays were correlated with antiviral activity. Figure 6a shows the outcome of several inhibitors against the HHV ANs from HSV (UL12), CMV (UL98) and KSHV (SOX). Compound 3, generated from docking studies (Figure 6b), showed good inhibition of all three ANs (Figure 6a). This inhibition was strongly correlated with antiviral activity against both HSV and CMV (Figure 6c). Similarly, the hydroxyquinoline carboxamide, 4 (Figure 5), was shown to have strong activity against all three nucleases that correlated well with its antiviral activity (not shown).

We also tested integrase inhibitors, BXA and two 8-hydroxyquinolines (HQs), from our library for inhibition of HSV/CMV in cell culture, as well as inhibition of purified TMID proteins [183]. The HIV integrase inhibitors showed extremely weak anti-HSV/CMV activity and no significant inhibition of HSV protein targets (AN and PolExo) [183]. On the other hand, BXA exhibited a noticeable antiviral effect against both viruses and strong inhibition against purified PolExo. The strongly antiviral HQ compounds were potent inhibitors of AN with no discernable activity against PolExo. HQ-166 also inhibited viral DNA synthesis and replication compartment formation [183]. Furthermore, HQ-166 did not show antiviral activity against other viruses (including adeno, chikungunya, dengue, influenza and zika), indicating specificity for herpesvirus replication (NIH). Thus, with leads against both AN and PolExo, we are well-positioned to further optimize against the two different HHV targets to validate these targets for antiviral drug development.

## 7. nsp14/nsp10 Proofreading Exonuclease of Coronaviruses

The ExoN domain of nsp14 is a magnesium-dependent exoribonuclease that acts in a 3′→5′ fashion on both ss- and dsRNA [86,96]. Nsp14 is highly conserved among all the CoVs [81], and the ExoN active site contains a conserved “DEDDh” motif that can coordinate two divalent Mg ions [81,86,184]. Figure 7 shows the alignment of the key residues that form the active sites of ExoN from the seven CoVs known to infect humans. The alignment shows 100% identity for the catalytic residues and 80% identity/99% homology for the remaining amino acids lining the active site. This high degree of conservation strongly suggests that it should be possible to broadly target the CoV exoribonuclease activity. Pharmacological ExoN inhibitors are needed to more precisely elucidate the role of ExoN and validate its potential to serve as a new antiviral target.

### Inhibition of ExoN

Biochemical and computational screenings have been performed to identify small molecule inhibitors of the proofreading activity. Not surprisingly, many of these leads possess well-recognized metal-binding pharmacophores (Figure 8). A fluorescence-based biochemical assay was used to screen for inhibition of nsp14/nsp10 activity with a commercial library of 5000 compounds [185]. Using this screen, several micromolar hits were identified, including patulin and aurintricarboxylic acid (ATA). Both compounds were shown to limit viral replication in cell culture without affecting cell viability; however, these leads did not produce synergistic activity when combined with remdesivir. The three salicylate moieties of ATA are well-disposed to chelate the active site magnesium ions. Interestingly, patulin contains a highly activated electrophilic olefin that could produce an irreversible mode of inhibition [185]. A docking-based approach predicted that both ATA and the dye pontacyl violet (PV6R) would interact strongly with the conserved acidic residues in the active site of ExoN [186]. To validate this result, Vero cells were infected with *SARS-CoV-2* and treated with ATA (500 mM) and PV6R (200 mM), leading to 66.6- and 25-fold reductions in the viral genome copy number, respectively.

Another more recent study used a computer-generated model of ExoN in complex with RNA to evaluate potential ExoN inhibitors [187]. This in silico approach identified several hits, including N-hydroxyglutarimide 5, an early HIV lead, and the naturally occurring salicylate isobavachalcone, both of which contain well-recognized metal-chelating functionality. As noted for patulin, electrophilic alkenes present in chalcone derivatives 6 and 7 may allow for covalent modification of the enzyme (Figure 8). Compounds 5-7 showed reasonable target potency (IC_50_ 17.43–21.99 μM) and were further investigated for antiviral activity against a related seasonal coronavirus, *HCoV-OC43* [187]. Interestingly, while these compounds did not show antiviral activity as single agents, they did demonstrate synergistic activity when tested in combination with remdesivir.

## 8. Conclusions

The scale of the current pandemic has underscored the need for more effective and safe antiviral agents for both recurring and emerging viral pathogens. The success of targeting viral nucleases characterized by a two-metal-ion-dependent active site, such as HIV integrase and influenza PA endonuclease, with potent and selective small molecule inhibitors suggests that similar viral enzymes could also be exploited for new antiviral drug discovery. In this review, we evaluated the potential of targeting viral exonucleases from both human herpesvirus and coronavirus pathogens and reviewed the underlying contribution these enzymes make in the viral life cycle. PolExo and ExoN are critical for proofreading and ensuring replication fidelity. Targeting this vital function is anticipated to induce lethal mutagenesis through catastrophic levels of mutations in progeny genomes and to provide potential synergy with mutagenic nucleoside analogues. From the standpoint of drug discovery, exonucleases have received almost no attention compared to the endonucleases, such as integrase and PA. Although attempts have been made to identify early lead matter through screening, the majority of identified hits are compounds that indiscriminately bind metals. We are focused on the use of structure-based methods to optimize drug-like compounds that can engage the metal-binding centers of the exonuclease active sites with high selectivity. We believe this approach holds significant value in developing next-generation antiviral drugs.

## Figures and Tables

**Figure 1 viruses-14-01557-f001:**
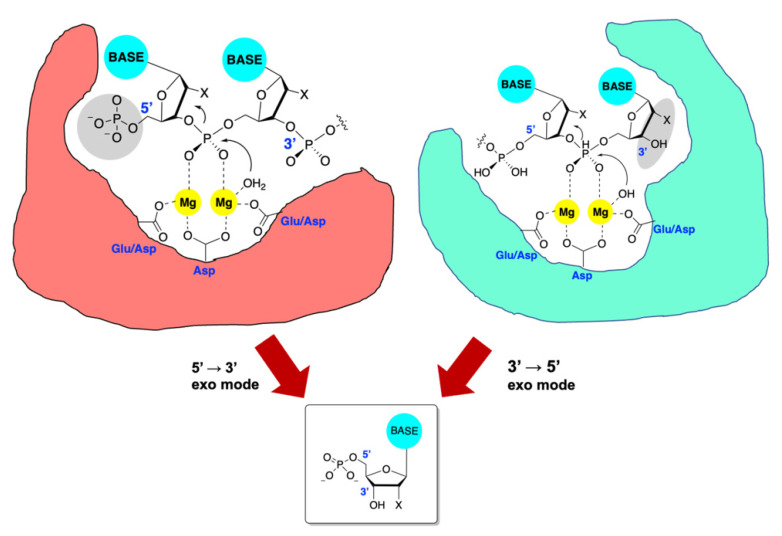
Exonuclease active sites have evolved to bind free 5′ or 3′ ends on DNA or RNA (but in a sequence-independent manner). The scissile phosphodiester is bound to the two divalent magnesium ions anchored by the DDE-type motifs, and an activated water molecule attacks the phosphorous atom in an S_N_^2^ displacement. Regardless of the manner of cleavage, a single ribo- or deoxyribonucleotide monophosphate is produced, generating a new 3′ or 5′ terminus for further cleavage. Important recognition domains are gray. Key recognition elements in grey, BASE = A, T, G, C or U, Mg = divalent magnesium, Glu = glutamic acid, Asp = aspartic acid, X = H or OH.

**Figure 2 viruses-14-01557-f002:**
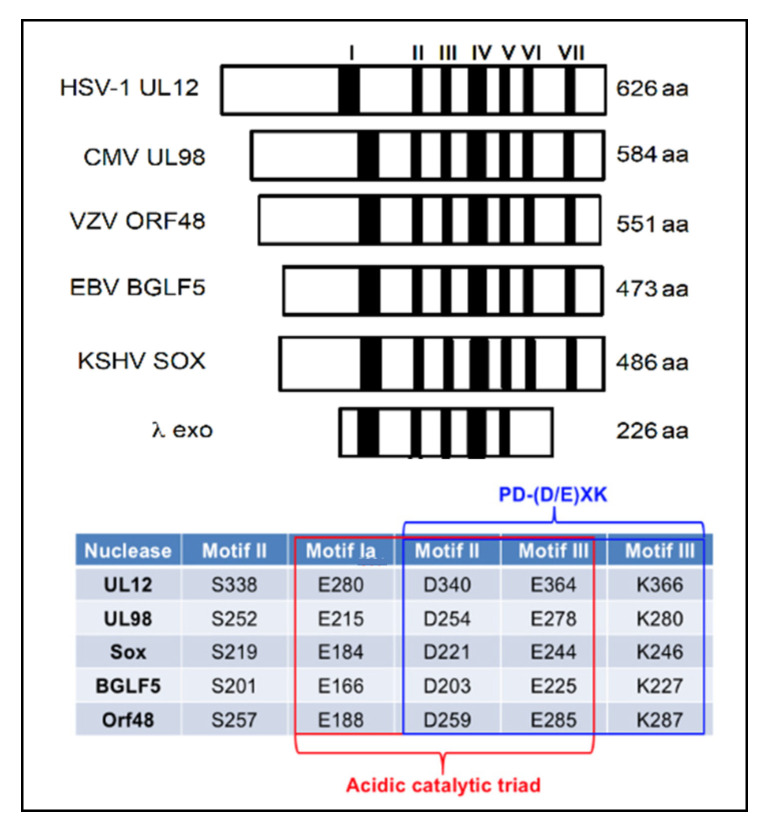
The seven conserved motifs of the herpesvirus alkaline nucleases as well as active site residues are shown. S = serine, E = glutamic acid, D = aspartic acid, K = lysine.

**Figure 3 viruses-14-01557-f003:**
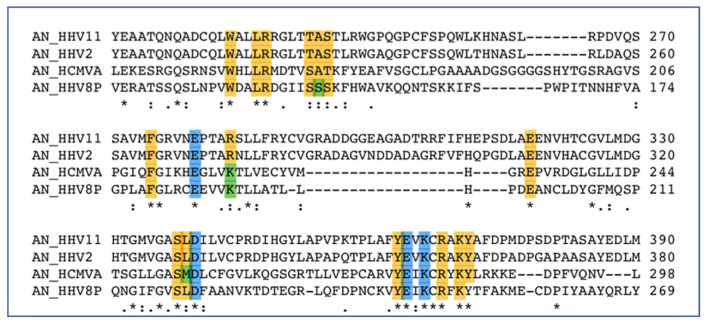
Active site sequence alignments of alkaline nucleases from HSV-1, HSV2, CMV and KSHV. Key metal binding EDEK residues highlighted in blue, other conserved active site residues not involved in metal binding are shown in yellow and green, where green indicates notable variations between family members. Asterisk denotes identical residues, colon denotes homologous residues and period denotes a low level of similarity.

**Figure 4 viruses-14-01557-f004:**
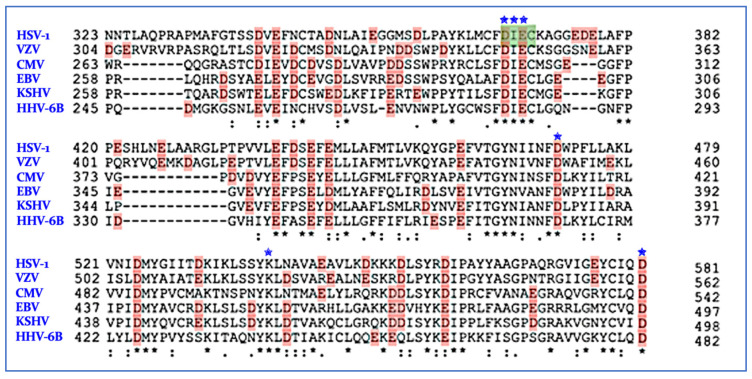
Active site sequence alignments of Pol/Exo regions of HSV-1, VZV, CMV, EBV, KSHV and HHV-6B. Virus name in blue text, key DIEDKD residues in red and denoted by ✬, other active site residues are in green, black stars denote identical residues, colon denotes homologous residues and period denotes a low level of similarity.

**Figure 5 viruses-14-01557-f005:**
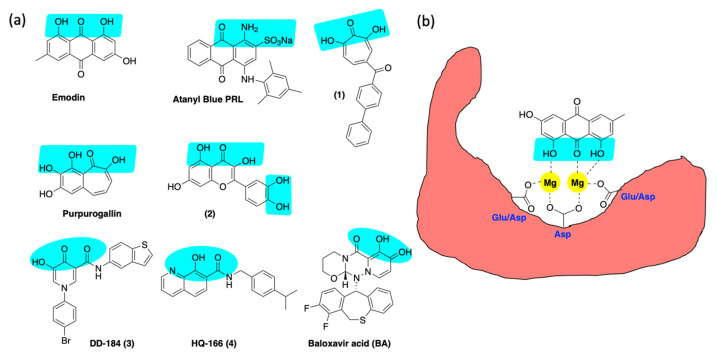
(**a**) Reported inhibitors of HSV exonucleases. Putative magnesium-binding functionality is highlighted in blue. (**b**) Hypothetical interactions between the inhibitor emodin and the TMID active site highlighting direct interactions between the anthraquinone oxygens and the bound magnesium ions.

**Figure 6 viruses-14-01557-f006:**
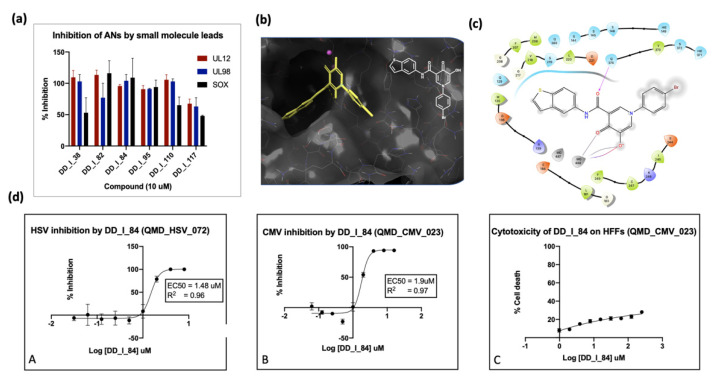
(**a**) Small molecule inhibition of HHV ANs from HSV, CMV and KSHV; (**b**) docking of compound 3 to SOX active site; (**c**) lig plot showing key interactions of the inhibitor and active site residues; (**d**) cytotoxicity and antiviral activity of compound 4 against HSV and CMV and relevant cytopathic effect against HFF cells.

**Figure 7 viruses-14-01557-f007:**
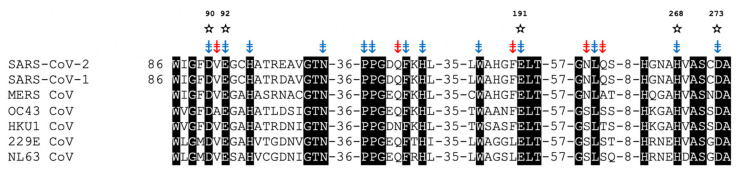
Sequence alignments of ExoNs from the 7 CoVs known to infect humans. ✫ = “DEDDh” motif; Active site residues; ⇟ = Identical, ⇟ = Highly Similar (polarity/size).

**Figure 8 viruses-14-01557-f008:**
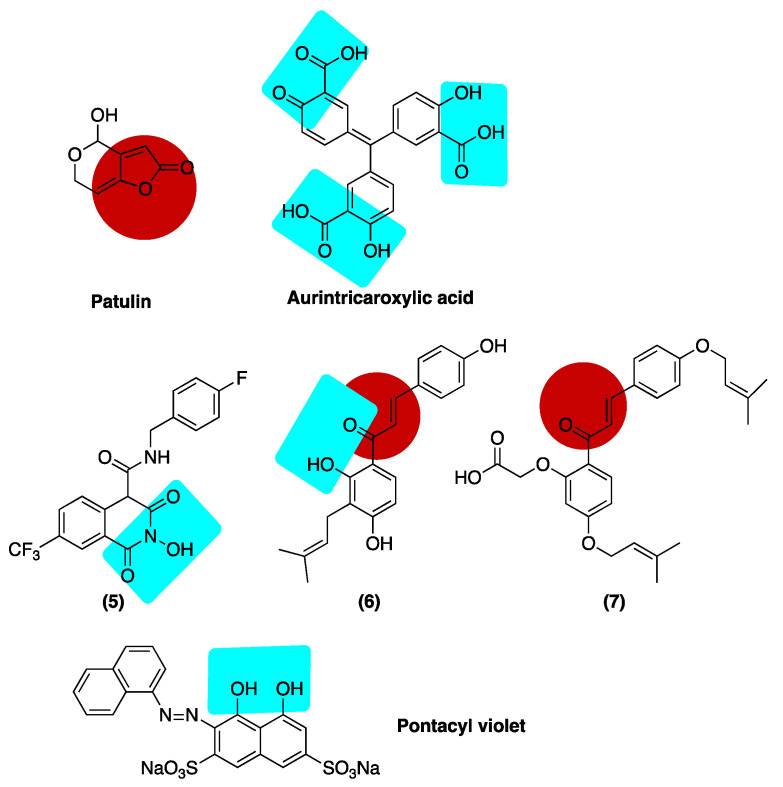
Inhibitors of the ExoN domain of nsp14 from SARS-CoV-2. Putative magnesium-binding functionality is highlighted in blue, while potential electrophiles are highlighted in red.

## Data Availability

Not applicable.

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
