# Peer review of "Viral Nucleases from Herpesviruses and Coronavirus in Recombination and Proofreading: Potential Targets for Antiviral Drug Discovery"

_viruses, 2022, doi:10.3390/v14071557_

Round 1

Reviewer 1 Report

Weller et. al reported "Viral nucleases from herpesviruses and coronavirus in recombination and proofreading; potential targets for antiviral drug discovery" interesting and well written. However manuscript needs minor revision as specified below. 

1) Some of the recent references regarding herps virus related drug materials based on chemical perspectives need to be added in brief specially in Current standard of care section. 

2) Herps and Corona virus antiviral drug mechanism schematic representation need to be provided in respective sections. 

3)  There are few typo's need to be corrected. 

4) Some of the figures not clearly visible such as Active site sequence (figure 4). 

Reviewer 2 Report

Article entitled “Viral nucleases from herpesviruses and coronavirus in recombination and proofreading; potential targets for antiviral drug discovery” explored the recombination in two very different virus families (Herpesviridae and Coronaviridae) that have become major threats to human health. This review focused on the roles of these three nucleases in their respective life cycles and discuss the state of drug discovery efforts against these targets. This article may be considered after revision.

Comments:

·      Line no 18-19 : only these two viruses evolved to rely heavily on recombination for viral genome replication, adaptation to new hosts and evasion of host immune regulation? Or Any other viruses ?

·      Detailed information is provided but adding/highlighting justification for , why only these viruses will give provide a clear understanding

·      Current standard of care for both virus: add drug lists either in table  form or figure (compound structure), discuss the mechanism as well   

·      Figure 5, 8 : provide protein-drug interaction figure in supplementary  if available

·      Figure 6: highlight the interacting residues

·      Line no 562-564 : structure-based methods?  ,   highlight in discussion
